# Dynamical behaviors and stability of bubbles and vortices in two-dimensional Bose quantum liquids

**Shiyi Wang[1], Liang Duan[1*], Liangwei Dong[2] and Zhan-Ying Yang[1,3,4]**

**1** School of Physics, Northwest University, Xi'an 710127, China
**2** Department of Physics, Zhejiang University of Science and Technology, Hangzhou, 310023, China
**3** Peng Huanwu Center for Fundamental Theory, Xi'an 710127, China
**4** Shaanxi Key Laboratory for Theoretical Physics Frontiers, Xi'an 710127, China

* liangduan1212@nwu.edu.cn

## Abstract

We investigated the structure and dynamical behavior of quantum bubbles and vortices in a two-dimensional uniform Bose quantum liquid. Through effective potential analysis and numerical calculations, we established the parameter regimes in which these two quantum states exist, finding that a narrow region below the equilibrium liquid density allows for their coexistence. In the coexistence region, when bubbles move relative to the background, increasing velocity induces a topological transition, manifested by the emergence of vortex-antivortex cores inside the bubble. Moving vortices appear as vortex-antivortex pairs, and in the entire coexistence region, as well as at slightly higher densities, these pairs exhibit an anomalous behavior in which the core separation first decreases with increasing velocity over a certain velocity range, a phenomenon markedly different from the predictions for Bose gases in mean-field theory. Finally, we analyzed the excitation energy-velocity relationship and, together with numerical time evolution, confirmed that both bubbles and vortices remain stable in the region where the excitation energy decreases as velocity increases. These findings provide new insights into the structure and stability of nonlinear excitations in quantum liquids.

# 1 Introduction

Ultradilute Bose quantum liquids, formed by quantum fluctuation–induced liquefaction of Bose-Einstein condensates (BECs), have attracted considerable attention [1–11]. In contrast to liquid helium, which is stabilized by strong classical van der Waals forces, a distinctive feature of the system is that the interatomic interactions can be tuned via Feshbach resonances, thereby allowing control over the properties of the liquid. This tunability renders Bose quantum liquids an excellent platform for exploring liquid-state physics and the role of quantum fluctuations in condensates. To date, most studies have focused on self-bound quantum droplets with finite atom numbers, addressing their collective excitation modes [12–17], inter-droplet interactions [18–20], modulational instabilities [21–23], and vortex droplets together with their stability [24–31]. The emergence of self-bound quantum droplets represents a direct manifestation of fluctuation-induced liquefaction in condensates, constituting one prominent mode of Bose quantum liquids.

Recently, a novel excitation distinct from self-bound structures has been proposed—termed the quantum bubble—which appears as a density defect embedded in the uniform liquid while maintaining a spatially homogeneous phase in the stationary case [32]. Research in one- and two-dimensional systems has established that bubbles exist within the density range bounded by the spinodal point and the equilibrium density, but stationary bubbles are always unstable, either expanding significantly or being filled in [33–36]. Recent studies have demonstrated the existence of stable traveling vortices in quantum liquids [36]. In contrast, the structural characteristics of traveling bubbles, as well as the question of whether they can remain stable, remain unresolved. On the other hand, our findings demonstrate that vortices can also exist in quantum liquids and, within a certain parameter regime, coexist with bubbles. As both bubbles and vortices represent defect excitations in the uniform liquid, a systematic study of their differences and common features may yield valuable insights into how quantum fluctuations influence the excitation mode of Bose-Einstein condensates.

In this paper, we study the structural and dynamical properties of bubbles and vortices in two-dimensional quantum liquids. Specifically, effective potential analysis and simulations reveal bubbles between the spinodal and equilibrium densities, and show that vortices appear when the background density exceeds a threshold below equilibrium. Notably, in the coexistence region of the two quantum states, traveling bubbles undergo a topological transition once their velocity exceeds a certain critical value, where a pair of vortex-antivortex cores emerges inside. The traveling vortex manifests as a vortex–antivortex pair, showing a marked distinction from bubbles at low velocities, while this difference vanishes at high velocities. Finally, by combining the analysis of excitation energy curves with dynamical evolution, we determine the velocity ranges over which traveling bubbles and vortices remain stable.

The structure of this paper is as follows: In Sec. 2, we reformulate the model into a Newtonian form and, through theoretical analysis of the effective potential at different densities, derive the existence conditions for bubbles and vortices. In Sec. 3, we first present the structural characteristics of bubbles and vortices obtained numerically, and then discuss the stability and dynamical properties of bubbles through numerical evolution. Finally, Sec. 4 summarizes our main findings and outlines possible directions for future research.

## 2 Existence regions of bubbles and vortices

We consider a Bose mixture composed of the $^{39}$K spin states $|\uparrow\rangle = |F, m_F\rangle = |1, -1\rangle$ and $|\downarrow\rangle = |1, 0\rangle$. As demonstrated in existing experiments, at an external magnetic field of about 56.9G, the intraspecies three-dimensional scattering lengths are $a_{\uparrow\uparrow}^{3D} \approx 37a_0, a_{\downarrow\downarrow}^{3D} \approx 85a_0$, while the interspecies scattering length is $a_{\uparrow\downarrow}^{3D} \approx -56a_0$ [5, 6], where $a_0$ is the Bohr radius. A tight harmonic trap is imposed along $z$ with oscillator length $l_z = 0.1\mu$m, satisfying $0 < -a_{\uparrow\downarrow}^{3D} < a^{3D} \ll l_z$ with $a^{3D} = \sqrt{a_{\uparrow\uparrow}^{3D} a_{\downarrow\downarrow}^{3D}}$, so that the system can be reduced to an effective two-dimensional geometry [2]. Our analysis is restricted to the common-mode ($n_\uparrow = \sqrt{g_{\downarrow\downarrow}/g_{\uparrow\uparrow}}n_\downarrow = \frac{\sqrt{g_{\downarrow\downarrow}}}{\sqrt{g_{\uparrow\uparrow}} + \sqrt{g_{\downarrow\downarrow}}}n$) configuration of the two components; in this case the energy density of a uniform system, including the LHY correction due to quantum fluctuations, is given by [2]:

$$\mathcal{E} = \frac{mg_{\uparrow\uparrow}g_{\downarrow\downarrow}}{8\pi\hbar^2}n^2 \ln\left[\frac{\sqrt{eg_{\uparrow\uparrow}g_{\downarrow\downarrow}}mn}{\Delta\hbar^2}\right], \tag{1}$$

where $m$ is atom mass, $n$ is uniform liquid density, $g_{\sigma\sigma} = \frac{4\pi\hbar^2}{m\ln[4e^{-2\gamma}/(a_{\sigma\sigma}^2\Delta)]}$ denotes the intraspecies coupling constant, $\Delta = \frac{4e^{-2\gamma}}{a_{\uparrow\downarrow}\sqrt{a_{\uparrow\uparrow}a_{\downarrow\downarrow}}}\exp\{-\frac{\ln^2(a_{\downarrow\downarrow}/a_{\uparrow\uparrow})}{2\ln[a_{\uparrow\downarrow}^2/(a_{\uparrow\uparrow}a_{\downarrow\downarrow})]}\}$, $\gamma$ is the Euler constant and $a_{\sigma\sigma'}$ is the two-dimensional scattering length between components $\sigma$ and $\sigma'$; the relation between the two- and three-dimensional scattering lengths satisfies

$$a_{\sigma\sigma'} = \left(\frac{4\pi}{\alpha}\right)^{1/2} l_z \exp\left[-\gamma - \sqrt{\frac{\pi}{2}}\frac{l_z}{a_{\sigma\sigma'}^{3D}}\right] (\sigma, \sigma' = \uparrow \text{ or } \downarrow), \tag{2}$$

where $\alpha \approx 0.905$ is a constant. The thermodynamic stability condition $\partial^2\mathcal{E}/\partial n^2 > 0$ implies that stable uniform liquids require $n > n_s$, where $n_s = \frac{\Delta\hbar^2}{e^2m\sqrt{g_{\uparrow\uparrow}g_{\downarrow\downarrow}}}$ is spinodal point density. The zero-pressure condition $P = \mu n - \mathcal{E} = 0$, where $\mu = \partial\mathcal{E}/\partial n$ is uniform liquid chemical potential, yields the equilibrium density $n_0 = \frac{\Delta\hbar^2}{e^{3/2}m\sqrt{g_{\uparrow\uparrow}g_{\downarrow\downarrow}}}$. When excitation modes are considered, the system becomes spatially inhomogeneous. The dynamics are then governed by the extended Gross-Pitaevskii equation (eGPE), which can be derived via variational minimization of the total energy [2]:

$$i\hbar\frac{\partial\Psi}{\partial T} = -\frac{\hbar^2}{2m}\nabla_R^2\Psi + \frac{mg_{\uparrow\uparrow}g_{\downarrow\downarrow}}{4\pi\hbar^2}|\Psi|^2\ln\left[\frac{|\Psi|^2}{\sqrt{e}n_0}\right]\Psi. \tag{3}$$

Using a trial wave function $\Psi$, the standard Bogoliubov-de Gennes (BdG) procedure yields the excitation spectrum [37]:

$$\epsilon(p) = \sqrt{\frac{p^2}{2m}\left[\frac{p^2}{2m} + \frac{mg_{\uparrow\uparrow}g_{\downarrow\downarrow}}{4\pi\hbar^2}n\ln\left(\frac{\sqrt{e}n}{n_0}\right)\right]}. \tag{4}$$

In the long-wavelength limit $p \to 0$, the spectrum becomes phonon-like, $\epsilon(p) = cp$, where $c = \sqrt{\frac{g_{\uparrow\uparrow}g_{\downarrow\downarrow}}{8\pi\hbar^2}n\ln\left(\frac{\sqrt{e}n}{n_0}\right)}$ is the speed of sound. The healing length is then defined as $\xi = \hbar/\sqrt{2}mc$, characterizing the spatial scale over which the condensate recovers from a perturbation [38].

To simplify analysis, we introduce dimensionless variables $(x, y) = (X, Y)/\xi_0$, $t = T/t_0$, $\psi = \Psi/\sqrt{n_0}$, where $\xi_0 = \sqrt{8\pi\hbar^4/(m^2 g_{\uparrow\uparrow}g_{\downarrow\downarrow}n_0)}$ is the healing length at the equilibrium density and $t_0 = m\xi_0^2/\hbar$, which leads to the dimensionless form of the eGPE:

$$i\frac{\partial\psi}{\partial t} = -\frac{1}{2}\nabla^2\psi + |\psi|^2\ln\left(\frac{|\psi|^2}{\sqrt{e}}\right)\psi, \tag{5}$$

84 where $\nabla^2 = \partial_{xx} + \partial_{yy}$. In this rescaled system, the characteristic parameters are given by:
85 $n_0 = 1, n_s = 1/\sqrt{e}$. For the excitation modes in a uniform system, taking into account the
86 rotational symmetry, we express the wave function as $\psi(r,\theta) = \phi(r)e^{iS\theta - i\mu t}$, where $\mu$ is the
87 dimensionless chemical potential, $S = 0, \pm1, \pm2, \ldots$ is the winding number, and $\phi(r)$ is a real
88 radial function satisfying

$$\mu\phi = -\frac{1}{2}\frac{d^2\phi}{dr^2} - \frac{1}{2r}\frac{d\phi}{dr} + \frac{S^2}{2r^2}\phi + \phi^3 \ln\left(\frac{\phi^2}{\sqrt{e}}\right). \tag{6}$$

89 For localized excitation modes on a homogeneous background, the boundary condition $\lim_{r\to\infty} \phi^2$
90 $= n_b$ holds, where $n_b$ is the background density, and the chemical potential is $\mu = n_b \ln(n_b/\sqrt{e})$.
91 By treating $r$ as an effective time and $\phi$ as the dynamical coordinate, Eq. (6) can be recast
92 into an effective Newtonian form, describing the motion of a unit-mass particle in a potential
93 $U(\phi)$ under additional forces [33, 34, 39]: $\frac{d^2\phi}{dr^2} = -\frac{dU}{d\phi} + F_D + F_S$, where the effective potential
94 is $U(\phi) = n_b \ln(n_b/\sqrt{e})\phi^2 - \frac{1}{2}\phi^4 \ln(\phi^2/e)$, $F_D = -(d\phi/dr)/r$ denotes the additional force
95 field arising from dimensional effects, and $F_S = \phi S^2/r^2$ denotes the additional force field due
96 to rotation. Analyzing the combined effect of the effective potential and the additional force
field helps us identify the excitation modes in the system. Initially, we ignore the influence of

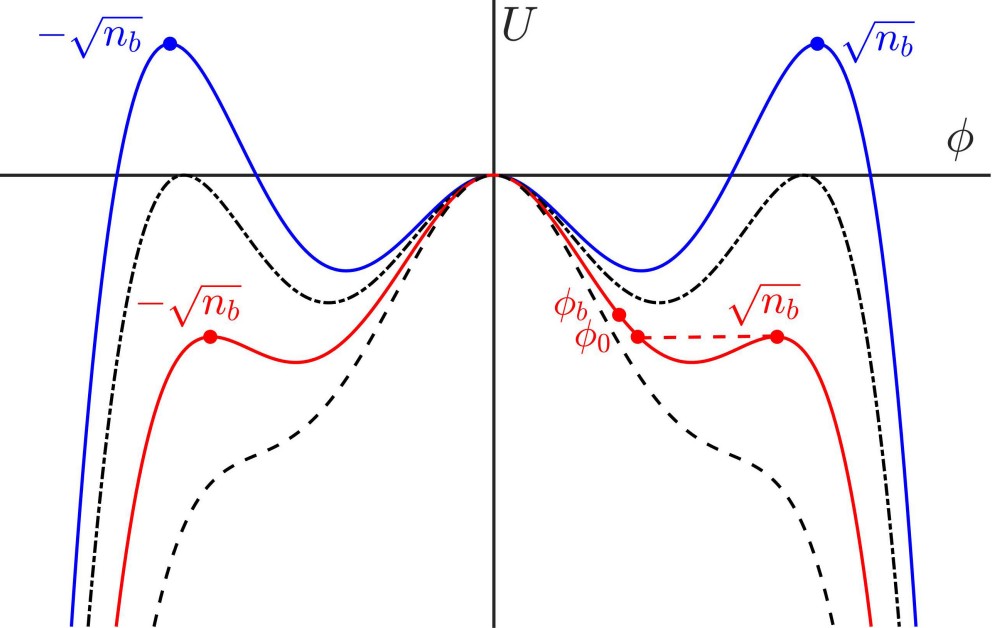

Figure 1: Effective potential $U(\phi)$ plotted for various values of the background density $n_b$. Four representative cases are shown: $n_b < n_s$ (black dashed), $n_s < n_b < n_0$ (red solid), $n_b = n_0$ (black dash dotted), $n_b > n_0$ (blue solid). These regimes correspond to unstable, bubble-supporting, marginal, and vortex-supporting configurations, respectively.

97
98 the additional force field and focus on the effective potential $U$; the effects of $F_D$ and $F_S$ will be
99 discussed later. For $n_b < n_s$, the thermodynamic analysis has already revealed that the system
100 is unstable, which is reflected in the effective potential $U$ by the absence of a confining well;
101 consequently, no stationary solution is expected to emerge, see the dashed curve in Fig. 1.
102 For $n_s < n_b < n_0$, the potential $U$ exhibits a double-well profile, featuring local maxima at
103 $\phi = \pm\sqrt{n_b}$ and a global maximum at $\phi = 0$. To facilitate analysis, we extend the domain of $r$
104 to $(-\infty, \infty)$. In this case, the potential $U$ permits the following configuration: as $r \to -\infty$,

$\phi = \sqrt{n_b}$; at $r = 0$, $\phi$ reaches a value $\phi_0$ satisfying $U(\phi_0) = U(\sqrt{n_b})$; and as $r \to \infty$, $\phi$ returns to $\sqrt{n_b}$. This corresponds to a localized bubble state, as shown by the red solid line in Fig. 1. At $n_b = n_0$, the potential has three global maxima at $\phi = 0$ and $\phi = \pm\sqrt{n_b}$. The potential allows a configuration where $\phi = \sqrt{n_b}$ at $r = -\infty$ and $\phi = 0$ at $r = \infty$, which corresponds to a kink solution in one dimension; however, such a configuration does not exist in our rotationally symmetric two-dimensional system. Still, from a limiting perspective, as $n_b \to n_0$, the $\phi_0$ lingers increasingly near 0, until at $n_b = n_0$ it corresponds to an infinitely extended bubble; see the black dashed line in Fig. 1. For $n_b > n_0$, the values $\phi = \pm\sqrt{n_b}$ become the global maxima of the potential. The potential permits a solution where $\phi$ evolves from $\sqrt{n_b}$ at $r = -\infty$ to $-\sqrt{n_b}$ at $r = \infty$. At $r = 0$, the wavefunction $\phi$ changes sign, producing a $\pi$ phase jump. In a one-dimensional system, this corresponds to a dark soliton, while in the rotationally symmetric two-dimensional system, it corresponds to a vortex solution, where any line crossing the vortex core diameter shows a $\pi$ phase change across the vortex core.

We now consider the role of the additional force field in the excitation modes. When $r$ changes from $-\infty$ to 0, the radial wavefunction $\phi$ decreases monotonically, resulting in $F_D < 0$, which facilitates the decrease of $\phi$. Hence, for bubble states $F_S = 0$, $\phi$ continues to decrease beyond $\phi_0$, indicating that for the same $n_b$, a bubble localized in both $x$ and $y$ directions exhibits a lower density compared to a line-like bubble localized only along $x$. To conclude, the bubble state exists for $n_s < n_b < n_0$. For vortex states, the dimensional force $F_D$, consistent with the previous analysis, promotes the decrease of $\phi$, which lowers the critical density for vortex excitation. In addition to $F_D$, one must account for the rotationally induced force $F_S$. When $r$ evolves from $-\infty$ to 0, $\phi$ decreases from $\sqrt{n_b}$. At this stage, since $F_S = \phi S^2 / r^2 > 0$, it behaves as a repulsive effect that resists the reduction of $\phi$. The critical density $n^*$ for vortex excitation is jointly determined by $F_D$ and $F_S$; this mechanism permits the coexistence of bubbles and vortices in the region $n^* < n_b < n_0$, a prediction confirmed by subsequent numerical results. The present study focuses on structural differences between vortices and bubbles; accordingly, in what follows, we restrict attention to the fundamental vortex with topological charge $S = 1$, as the results for higher charges are analogous to those of the fundamental case.

## 3 Structure, Stability, and Dynamics of Bubbles and Vortices

Above, we theoretically analyzed the density regions in which the two quantum states can exist. In this section, we present numerical results on the structure, stability, and dynamical properties of bubbles and vortices. In our subsequent numerical simulations, the spatial grid size is set to $L_x = L_y = 100$ with $1000^2$ discrete points, and periodic boundary conditions are applied. For the dynamical evolution, the time step is $dt = 0.001$. Under the experimental parameters chosen above, $\xi_0 \approx 4\mu m$, $t_0 \approx 8ms$, and $n_0 \approx 4\times 10^9 cm^{-2}$), the overall condensate extends over a spatial scale of approximately $400\mu m$, with the evolution time on the order of seconds.

### 3.1 Stationary bubbles and vortices

As a starting point, and in line with the theoretical analysis above, we examine the case of excitations at rest with respect to the background. By setting $S = 0$ and $S = 1$ in Eq. (6), the bubble and vortex states can be obtained via the squared-operator iteration method. The density profiles of the bubble at different values of $n_b$ are illustrated in Fig. 2(a). The results indicate that the bubble vanishes as $n_b \to n_s$; as $n_b$ increases, the minimum density of the bubble decreases, and a prominent flat-bottom structure forms as $n_b \to n_0$. To provide a more precise description of the bubble structure, we show in Fig. 2(b) how the minimum density of

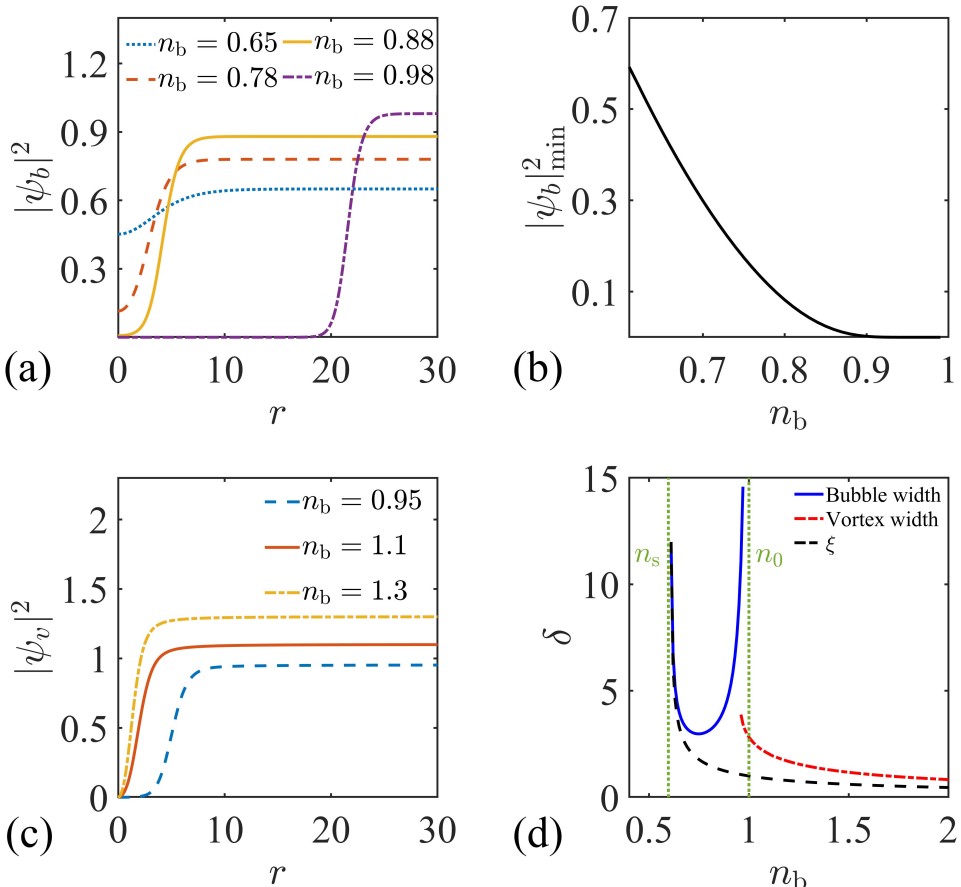

Figure 2: Stationary density profiles and characteristic parameters of bubbles and vortices at various $n_{\mathrm{b}}$. (a) Radial density distributions of stationary bubble states for $n_{\mathrm{b}} = 0.65$ (blue dotted), $n_{\mathrm{b}} = 0.78$ (orange dashed), $n_{\mathrm{b}} = 0.88$ (yellow solid), and $n_{\mathrm{b}} = 0.98$ (purple dash-dotted). (b) Minimum density at the bubble center as a function of $n_{\mathrm{b}}$. (c) Radial density distributions of vortices for $n_{\mathrm{b}} = 0.95$ (blue dashed), $n_{\mathrm{b}} = 1.1$ (orange solid), and $n_{\mathrm{b}} = 1.3$ (yellow dash-dotted). (d) Width of bubbles and vortex cores, along with the corresponding healing length, plotted as functions of $n_{\mathrm{b}}$.

the bubble varies with $n_{\mathrm{b}}$. It can be seen that within the existence regime of the bubble, the minimum density decreases with increasing $n_{\mathrm{b}}$. It is important to note that, unlike quantum vortices, the bubble is a non-topological excitation whose minimum density approaches but never exactly reaches zero. As both bubbles and vortices represent defect-like excitations on a uniform background, we define the width $\delta$ of the quantum state as the distance between the point of maximal density depletion and the location where the density reaches half the background value, namely, $(|\psi(\infty)|^2 - |\psi(0)|^2)/2 = |\psi(\delta)|^2 - |\psi(0)|^2$. The results are shown in Fig. 2(d), where we find that the bubble width first decreases and then increases with increasing $n_{\mathrm{b}}$. Although the bubble width diverges at both $n_{\mathrm{s}}$ and $n_0$, the underlying mechanisms are different. As previously introduced, the healing length of a condensate quantifies the distance over which density disturbances return to the background value. The healing length diverges at $n_{\mathrm{b}} = n_{\mathrm{s}}$, while remaining finite at $n_{\mathrm{b}} = n_0$. Thus, at $n_{\mathrm{b}} = n_{\mathrm{s}}$, the divergence of the bubble width as a defect excitation is rooted in the intrinsic properties of the condensate. However, at $n_{\mathrm{b}} = n_0$, the quantum bubble exhibits a large flat-bottom structure that far exceeds the spatial scale of density recovery; the healing length thus fails to properly characterize the bubble's

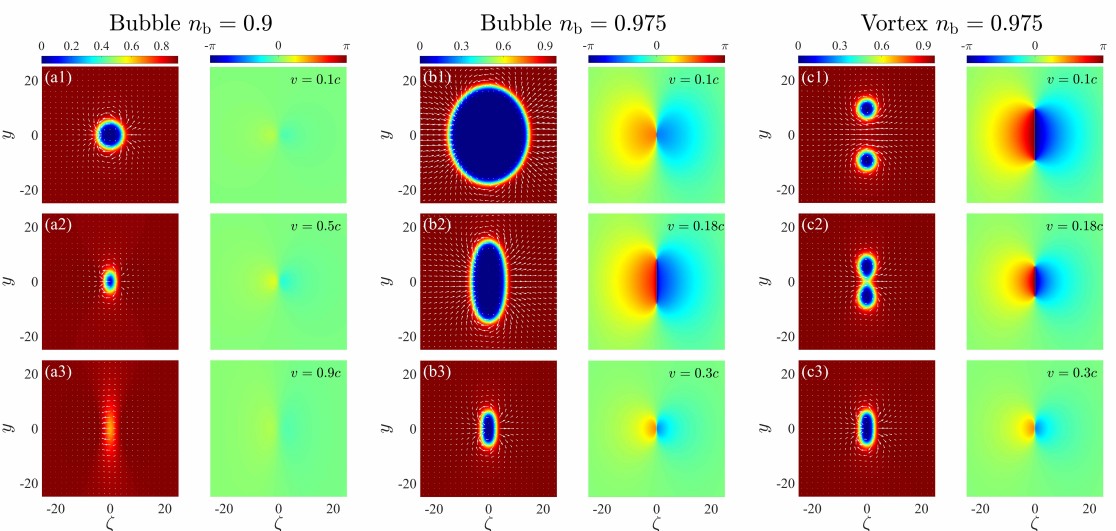

Figure 3: Panels (a1)–(a3) display density and phase patterns of traveling bubble with $v = 0.1c$, $0.5c$, $0.9c$ and $n_b = 0.9$. Panels (b1)–(b3) display density and phase patterns of bubble with $v = 0.1c$, $0.18c$, $0.3c$ and $n_b = 0.975$. Panels (c1)–(c3) display density and phase patterns of vortex dipoles with $v = 0.1c$, $0.18c$, $0.3c$ and $n_b = 0.975$.

width.

Fig. 2(c) shows the vortex structures at different $n_b$, in agreement with our theoretical expectations; for $n_b < n_0$, vortices and bubbles coexist, and near the critical density $n^* \approx 0.95$, a pronounced flat-bottom structure emerges. The vortex width is also shown in Fig. 2(d), where we find that at $n^*$, the vortex exhibits a divergent width while the healing length remains finite, which is due to the formation of a large flat-bottom structure making the healing length inadequate to characterize it. At high $n_b$, the vortex rapidly recovers from the core to the background, and the vortex width becomes proportional to the healing length.

## 3.2 Traveling bubbles and vortices

The above results demonstrate that the liquid density has a significant influence on the structures of bubbles and vortices. As both are defect-type excitations on a homogeneous background, their motion relative to the background breaks the isotropy of the kinetic energy due to spatially varying velocities, thereby also influencing their structure. Owing to the system's rotational symmetry, we may, without loss of generality, assume that the quantum state propagates along the $x$-direction to the background field. To facilitate the solution, we introduce a moving coordinate $\zeta = x - vt$. The wave function can be written as $\psi(x, y, t) = \phi(\zeta, y)e^{-i\mu t}$, where $\phi$ satisfies the equation

$$-iv\frac{\partial \phi}{\partial \zeta} + \mu\phi = \left[-\frac{1}{2}\nabla_\zeta^2 + \ln\left(\frac{|\phi|^2}{\sqrt{e}}\right)|\phi|^2\right]\phi, \tag{7}$$

where $\nabla_\zeta^2 = \frac{\partial^2}{\partial \zeta^2} + \frac{\partial^2}{\partial y^2}$. By solving Eq. (7) using the squared-operator iteration method, we obtain traveling bubble and vortex solutions.

We first consider the regime where only the bubble excitation is present, taking the $n_b = 0.9$ as an example (see Fig. 3(a)). With increasing velocity, the bubble experiences anisotropic compression. As the velocity approaches the speed of sound of the condensate, the density defect gradually vanishes, with the bubble becoming strongly flattened along the direction

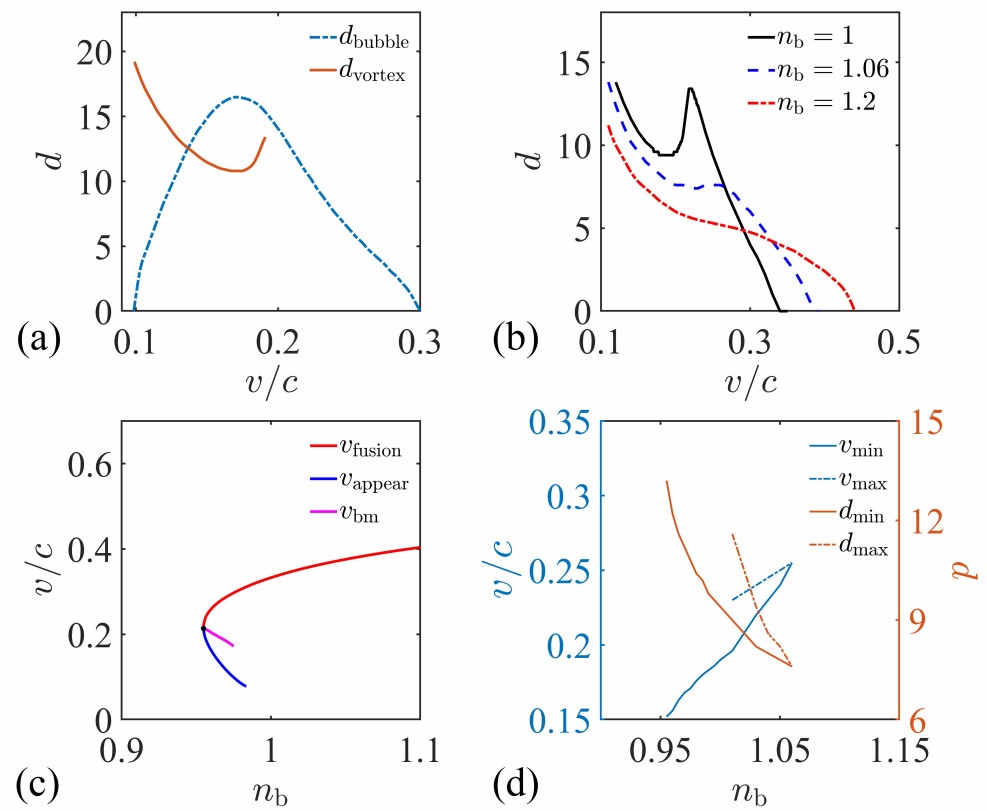

Figure 4: Velocity-dependent core dynamics of traveling bubble states and vortex dipoles. (a) The dependence of vortex-core spacing on velocity for vortex cores within traveling bubbles (blue) and for traveling vortices (orange) at $n_{\rm b} = 0.975$. (b) Vortex core separation versus velocity for vortex dipoles at three different values of $n_{\rm b}$. (c) Critical velocities associated with vortex structure transitions as functions of $n_{\rm b}$: velocity at which vortex dipoles annihilate (red), onset velocity for vortex-core nucleation in traveling bubbles (blue), and velocity corresponding to the maximal core separation in traveling bubbles (magenta). (d) Characterization of the anomalous non-monotonic dependence of vortex dipole core separation on velocity. The horizontal axis denotes $n_{\rm b}$. The left vertical axis (blue) indicates the velocity corresponding to the local minimum (solid) and local maximum (dashed) of the core separation. The right vertical axis (orange) shows the corresponding core separations at these extrema (solid and dashed, respectively).

189  of motion and stretched in the transverse direction. The white arrows depict the condensate
190  current field, given by $\mathbf{j} = |\phi|^2 \nabla \arg(\phi)$. Together with the phase distribution, it becomes
191  clear that the formation of oppositely circulating vortical flows on both sides of the bubble in
192  the direction perpendicular to motion, which gradually disappear as the velocity approaches
193  the speed of sound in the condensate. We next turn to the comparison of bubble and vortex
194  structures in the coexistence region, and we choose a representative background density of
195  $n_{\rm b} = 0.975$. As shown in the left column of Fig. 3(b), the bubble experiences progressive
196  compression with increasing velocity, and the density defect ultimately disappears as the ve-
197  locity nears the condensate sound speed, consistent with the low background density case (not
198  shown at high velocity). However, inspection of the phase distribution reveals that as the ve-
199  locity increases, a pair of oppositely charged vortex cores—a topological structure— emerges
200  at the bubble center once a critical velocity $v_{\rm appear}$ is reached. These vortex cores drift apart

as the velocity increases further. The vortex cores reach their maximum separation at velocity $v_{bm}$, after which the cores begin to approach each other and eventually annihilate at another critical velocity $v_{fusion}$. The dependence of the critical velocities $v_{appear}, v_{bm}, v_{fusion}$ on the background density is illustrated in Fig. 4(c). As the background density decreases, $v_{appear}$ and $v_{bm}$ exhibit an increasing trend, whereas $v_{fusion}$ decreases correspondingly. The three critical velocities converge at $n_b = 0.9548, v = 0.2136c$. For all background densities below this point, the bubble fails to produce topological charge.

The traveling vortex states, presented in Fig. 3(c), exhibit vortex-antivortex pair structures, in agreement with the theoretical framework proposed by Jones et al. As the velocity increases, the distance between the vortex pair decreases, and at a critical velocity $v_{fusion}$, the vortex and antivortex cores annihilate, after which the structural distinction between bubbles and vortices disappears. Fig. 4(a) shows the velocity dependence of the separation between vortex-antivortex cores and that between the vortex cores generated inside bubbles. Unlike the bubble case, the distance between the vortex and antivortex cores decreases, then increases, and eventually decreases again with increasing velocity until annihilation, which is in sharp contrast to the monotonic decrease predicted by mean-field theory. Therefore, we have

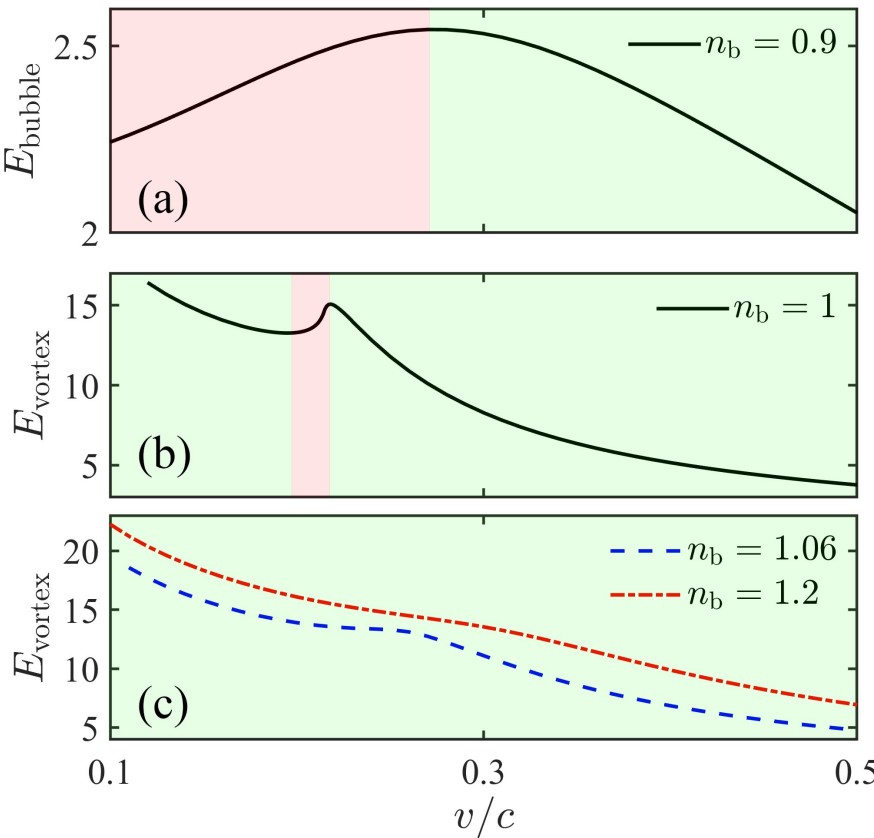

Figure 5: Excitation energy as a function of velocity for two types of traveling excitations. (a) Excitation energy of traveling bubble states at $n_b = 0.9$. (b)(c) Excitation energy of traveling vortices for $n_b = 1$(black solid), $n_b = 1.06$(blue dashed), and $n_b = 1.2$(red dash-dotted).Red areas correspond to unstable regimes, and green areas correspond to stable regimes.

further examined traveling vortices at different background densities, focusing on the relation between vortex core spacing and velocity, as shown in Fig. 4(b). The results indicate that once $n_b$ exceeds a certain critical value, the distance between vortex cores decreases monotonically

with increasing velocity. The dependence of the local minimum distance $d_{\text{min}}$ between vortex
pairs and the corresponding velocity $v_{\text{min}}$, as well as the local maximum distance $d_{\text{max}}$ between
vortex pairs and the corresponding velocity $v_{\text{max}}$, during the coalescence process, is displayed
in Fig. 4(d) as a function of $n_{\text{b}}$. It can be observed that the velocities corresponding to reaching
the extremal points increase with increasing $n_{\text{b}}$, while $d_{\text{max}}$ and $d_{\text{min}}$ all decrease with increas-
ing $n_{\text{b}}$. At $n_{\text{b}} \approx 1.06$, $d_{\text{max}}$ and $d_{\text{min}}$ coincide, and for $n_{\text{b}} > 1.06$, the distance between vortex
cores decreases monotonically with increasing velocity.

## 3.3 Dynamical characteristics

After addressing the structural features of quantum bubbles and vortices in homogeneous
quantum liquids, we next focus on their dynamical behavior. We begin by analyzing the dis-

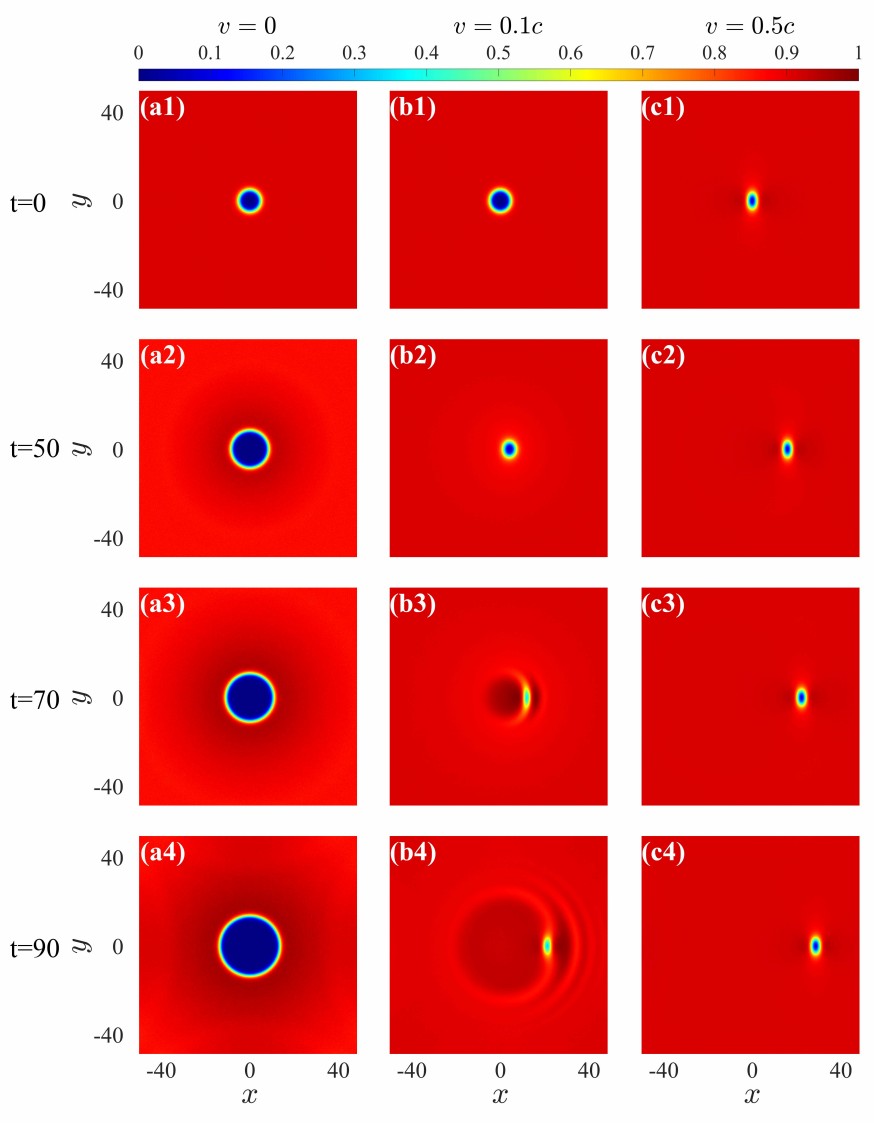

Figure 6: Dynamical evolution of bubble density profiles at $n_{\text{b}} = 0.9$, shown as snap-
shots at different times (see legend). (a) $v = 0$. (b) $v = 0.1c$. (c) $v = 0.5c$.

persion relations of bubbles and vortices, where for an excited state on top of the background,

the excitation energy is defined as [37, 38]:

$$E = \Omega - \Omega_G$$

$$= \iint \left[ \frac{1}{2}|\nabla\psi|^2 + \frac{1}{2}|\psi|^4 \ln\left(|\psi|^2/e\right) - \mu|\psi|^2 \right] dxdy$$

$$- \iint \left[ \frac{1}{2}n_b^2 \ln\left(n_b/e\right) - \mu n_b \right] dxdy, \tag{8}$$

where $\Omega$ is the grand potential of the excited state and $\Omega_G$ is the grand potential of the homogeneous system under the same background density. The results are presented in Fig. 5. For bubbles, we find that in all parameter regions where bubbles exist, the dispersion relation exhibits a non-monotonic behavior: the excitation energy first increases and then decreases with velocity, as illustrated in the example with $n_b = 0.9$. For traveling vortices, comparison with Fig. 4(b) shows that in the anomalous region—where the core-to-core distance grows as velocity increases—the excitation energy rises with velocity increases, while in the remaining regions it decreases as velocity grows.

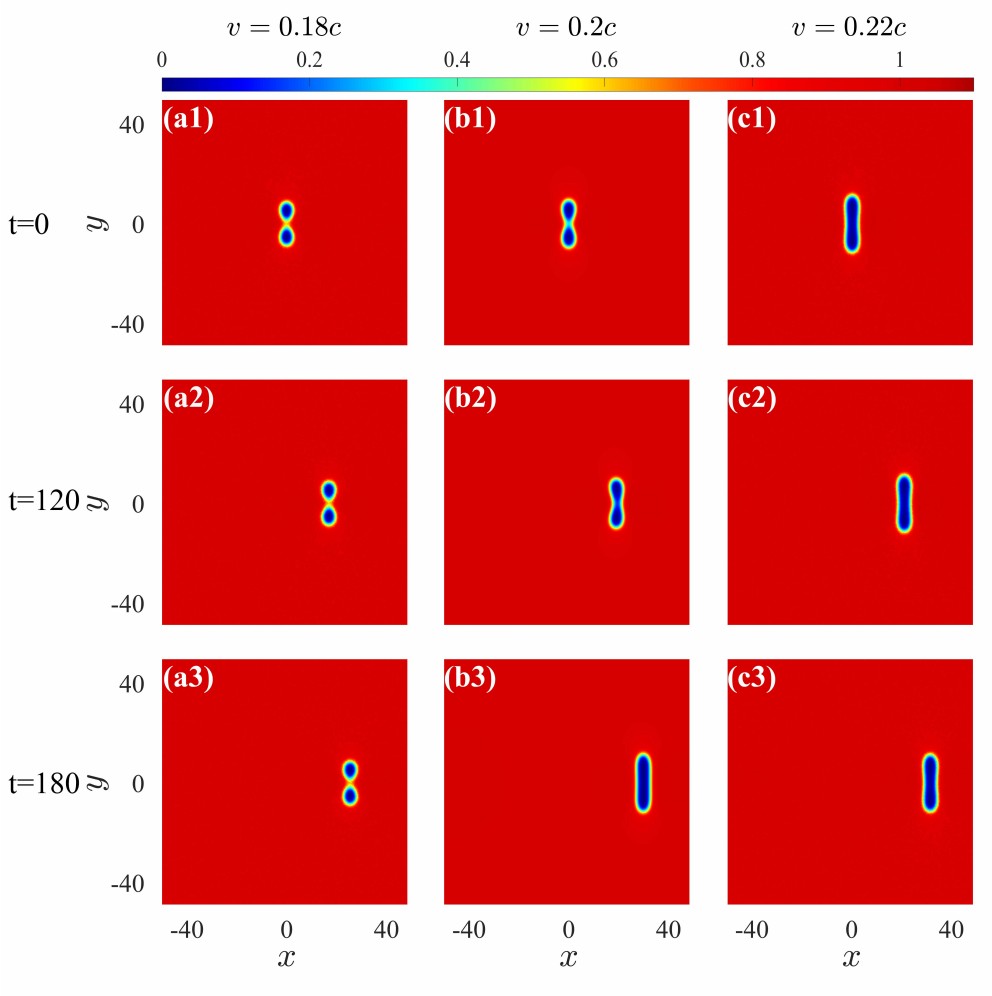

Figure 7: Dynamical evolution of traveling vortex density profiles at $n_b = 1$, shown as snapshots at different times (see legend). (a) $v = 0.18c$. (b) $v = 0.2c$. (c) $v = 0.22c$.

Next, we perform numerical simulations of Eq. (5) using the split-operator technique to investigate the dynamical evolution of the associated quantum states. The initial states are

taken as the obtained solutions with an additional 3% random perturbation. The results indicate that stationary bubbles are always unstable across the entire density range where bubbles can be excited. As an example, Fig. 6(a) shows the snapshots of a bubble at $n_b = 0.9$ at different times, revealing that the bubble structure is unstable and expands indefinitely, similar to the instability observed in one-dimensional bubbles. Next, we examine the case of a traveling bubble. Fig. 6(b) presents the density distribution of a bubble at different times for a velocity of $v = 0.1c$. It is observed that the bubble undergoes compression, followed by a breakdown of its structure, and eventually transforms into a faster-traveling bubble. This implies that bubbles with higher velocities could exhibit greater stability. As an example, we consider a bubble

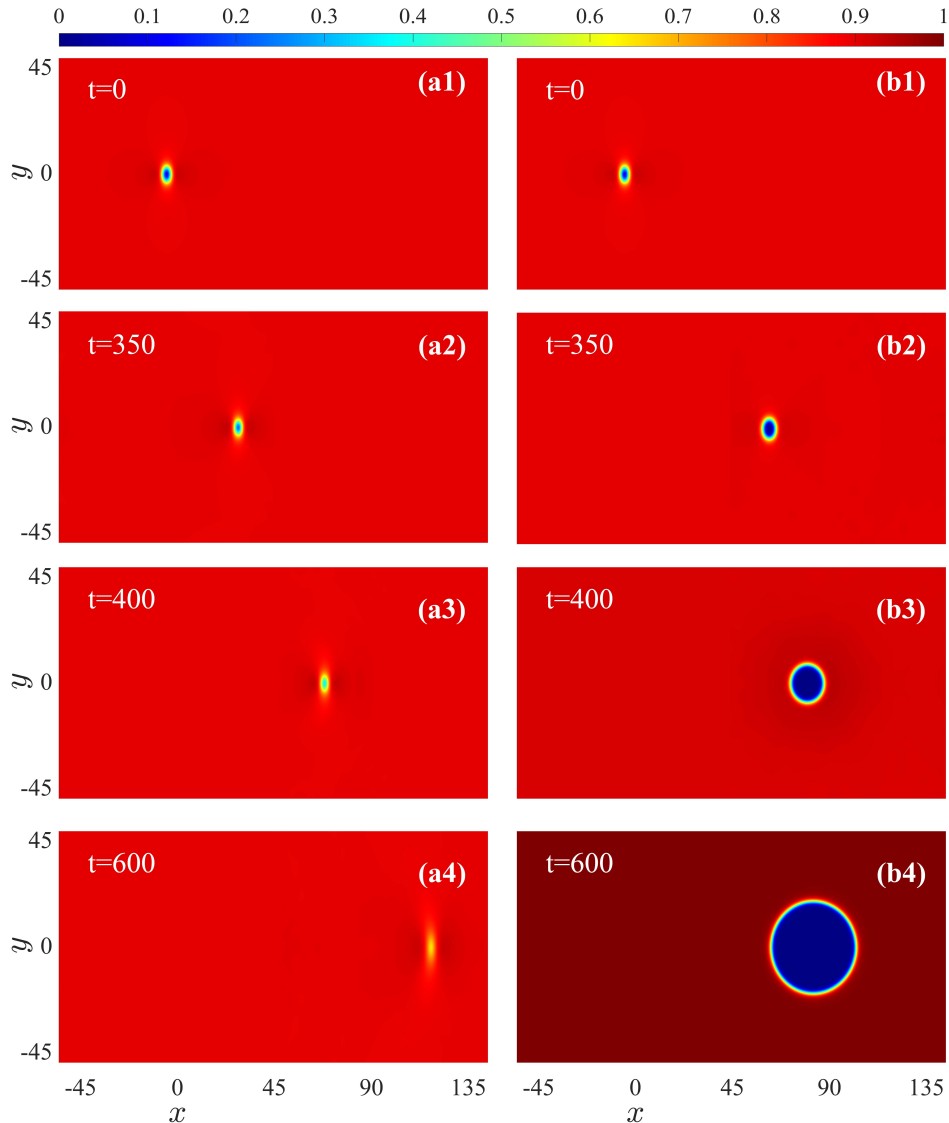

Figure 8: Driving traveling bubble by coupling to an impurity atom under external forcing, with initial conditions $n_b = 0.9$, $v = 0.5c$. (a) During acceleration, the bubble evolves smoothly and retains a structure consistent with the traveling-bubble solution. (b) During deceleration, the bubble becomes unstable and undergoes core expansion.

at $v = 0.5c$. Its density evolution at different times is illustrated in Fig. 6(c). It is evident that at this velocity, the bubble can preserve its structure well, suggesting that bubbles become

more stable at higher velocities. Considering vortex states with $n_b = 1$, we select $v = 0.18c$ and $v = 0.22c$ from the regime where excitation energy decreases with velocity, and $v = 0.2c$ from the regime where it increases. Fig. 7 presents the corresponding density snapshots at various times. The results show that the structures at $v = 0.18c$ and $v = 0.22c$ are preserved, while the one at $v = 0.2c$ experiences a splitting process during time evolution.

These results seem to indicate that both quantum states remain stable in the regime where the excitation energy decreases with increasing velocity. Therefore, we consider driving a stable state into both the unstable and stable regimes to investigate its structural changes, using the following setup: Following the idea proposed in Ref. [40–42], we couple the bubble to a small number of impurity atoms and apply an external force to the impurities, thereby driving the bubble through their mutual interaction. This setup transforms the system into a two-component model:

$$i\frac{\partial \psi}{\partial t} = -\frac{1}{2}\nabla^2\psi + |\psi|^2 \ln\left(\frac{|\psi|^2}{\sqrt{e}}\right)\psi + |\psi_i|^2\psi,$$
$$i\frac{\partial \psi_i}{\partial t} = -\frac{1}{2}\nabla^2\psi_i + |\psi|^2\psi_i + V\psi_i, \tag{9}$$

where $\psi_i$ denotes the impurity wave function, with $|\psi_i|^2 \ll 1$, and the potential $V = -Fx$ represents a linear potential, corresponding to a constant external force. We first consider the acceleration case by setting $F = -0.03$, since the initial state is in the regime where the excitation energy decreases with increasing velocity, corresponding to an effective negative mass. Thus, a negative $F$ is chosen. The time evolution of the system is shown in Fig. 8(a), where it is evident that the bubble maintains its structural integrity during the acceleration process. Conversely, when $F = 0.03$, the bubble is decelerated under the applied force. As shown in Fig. 8(b), the bubble becomes unstable as its velocity decreases, eventually leading to the complete breakdown of its structure. We performed simulations for a wider range of parameters and found the same results in all cases. Therefore, we conclude that both quantum states remain stable within the regime where the excitation energy decreases with increasing velocity.

## 4 Conclusion

We have investigated the structure and dynamics of bubbles and vortex pairs in two-dimensional Bose quantum liquids. The system is described by the eGPE with LHY corrections. The traveling vortex manifests as a vortex–antivortex pair, showing a marked distinction from bubbles at low velocities, while this difference vanishes at high velocities. In the region where the two quantum states coexist, we found that when the velocity of a traveling bubble exceeds a critical value, its topological properties change, manifested by the formation of vortex-antivortex cores inside the bubble, which, as the velocity increases, first move apart, then approach each other, and finally annihilate. We compared traveling vortices under the same parameters, finding that at low speeds their configuration is entirely different; as the velocity increases, the vortex-antivortex spacing first decreases, then increases, and finally decreases again, and after annihilation, the structure becomes indistinguishable from that of the bubble. In conventional Bose gases within the mean-field framework, the vortex-antivortex spacing varies monotonically with velocity, highlighting this as a unique property of vortex excitations in quantum liquids. Finally, through the combined study of excitation energy curves and dynamical evolution, we confirmed that both traveling bubbles and traveling vortices remain stable in the region where the excitation energy decreases with increasing velocity. These results provide guidance for experimental observations as well as for future studies of interactions between

bubbles, between bubbles and vortices, and beyond.

It is worth noting that our analysis focuses on excitations in a uniform liquid under the common-mode approximation; in finite-size systems, however, boundary effects become significant, and studying their impact on these excitations will further facilitate experimental observation. Moreover, Bose quantum liquids are intrinsically two-component systems. Beyond the common-mode approximation, the two components may support even richer structures, and the existence conditions, configurations, and dynamics of these quantum states remain to be investigated.

# Acknowledgements

**Funding information**   Financial support from the National Natural Science Foundation of China (Grant Nos. 12275213, 12247103, 12405003, 62575264).

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
