# Peer review of "Dynamical behaviors and stability of bubbles and vortices in two-dimensional Bose quantum liquids"

_SciPost Physics_

## Round 1 · Referee Report · Anonymous (Referee 1) · 2025-11-27

Strengths

1 - Interesting topic 2- Well written

Weaknesses

1 - Part of the results not original 2 - Not sure technical approach for numerical solutions is correct 3 - Goal of the paper cannot be related to groundbreaking progress 4- Contextualization and reference list not adequate

Report

The authors study dark excitations (bubbles, vortices and vortex dipoles) within infinitely large symmetric quantum droplets, namely they analyse the equation given in Eq. (5). The topic is interesting and, in fact, this equation has been studied in a number of relevant papers in the past. However, in my opinion, the paper fails to meet the high standard needed for publication in Scipost, for the reasons that I now detail.

First, part of the findings are very similar to previously published results. For instance, figure 1.a seems equivalent to figure 1 in reference [36]. In fact, panels a and c of figure 3, panels c of figure 6 and figure 7 are also very similar to the results of [36]. On the other hand, figure 6 panel b seems to show that the initial bubble is not an eigenstate. Broadly, I would say that ref [36] deals with the same questions in the same model, in particular the n_b=1 case in the present notation. Although it has some interest to generalize those results for different densities, that does not seem like a breakthrough development suitable for Scipost Physics. Moreover, [36] is not the only reference that deals with similar questions and the authors fail to quote important references and to relate their results to those previous discussions. This starts with C A Jones and P H Roberts 1982 J. Phys. A: Math. Gen. 15 2599. Curiously, in line 209, the authors mention Jones et al. but do not include a reference.

In this context, one of the motivations in the introduction: “Recent studies have demonstrated the existence of stable traveling vortices in quantum liquids [36]. In contrast, the structural characteristics of traveling bubbles, as well as the question of whether they can remain stable, remain unresolved” is surprising. Jones and Roberts demonstrated the existence of stable travelling waves without vorticity (rarefaction pulses) in two-dimensional NLSE, and this was generalised in [36] to the model at hand. Then, the results of Fig. 5 seem to be at odds with [36] and Jones-Roberts, with a non-trivial turnaround in the E vs. v curve, which is also argued to be related to a turnaround on how the vortices approach each other. Of course, [36] could be wrong and this paper could be right, but if that is the case, it should be clearly stated and the numerical procedure to find this new result clearly explained. The difference with the qualitative results of Jones-Roberts should be emphasised.

In summary, referencing and contextualization are improvable, the originality of all the results is not completely clear and there are technical details that also are not clear to me. Even if all these problems were addressed, this contribution does not meet the expectation of Scipost Physics. It is stated that one of the following conditions should be fulfilled for acceptance:

  • Provide a novel and synergetic link between different research areas.
  • Open a new pathway in an existing or a new research direction, with clear potential for multi-pronged follow-up work
  • Detail a groundbreaking theoretical/experimental/computational discovery
  • Present a breakthrough on a previously-identified and long-standing research stumbling block

This project is incremental work that might merit publication in some form, if amended. But I do not see how it could meet any of these four conditions.

In view of all this, I cannot recommend the contribution for publication in Scipost Physics.

Requested changes

I cannot envisage specific changes that may lead to acceptance.

Recommendation

Reject

---

## Round 1 · Referee Report · Anonymous (Referee 2) · 2025-11-29

Report

The manuscript of Wang and colleagues studies theoretically an effective two-dimensional model describing a binary Bose-Einstein condensate in the beyond-mean-field regime which has been an ongoing focus for the quantum gas community since the experimental observation of stable quantum droplets in weakly interacting dipolar and Bose mixtures. This has in turn facilitated a great interest in these unusual liquid-like states of matter, and the phenomenology of their associated ground and excited states.

The present work is a mostly numerical study of so-called bubbles [first predicted in this context by Naidon et al., Phys. Rev. Lett. 126, 115301 (2021) & Ref. 32] and quantum vortices, detailing the specific criterion for their existence as different physical parameters are varied, revealing regimes of stability and instability in the beyond-mean-field model for these excitations. After briefly discussing the individual solutions, the behaviour of pairs of these excitations is described (partially) in terms of the core’s of the solutions. Following this, the manuscript describes the dynamical properties of the bubbles and vortices under different initial conditions in the presence of a random perturbation. In the final part of their work the authors examine the coupling of an impurity to the bubble solution, exploring the effect of a driving force applied to the impurity component.

While the manuscript is overall well-written and the authors’ results provide some new insight into this highly active field, the work in several places could benefit from being more detailed and quantitative. As such, there are quite a few questions and concerns that should be addressed and clarified for a potential recommendation for publication. If the authors resubmit a revised version of their work, they should fully respond to the following points in their response:

(i) Figure three presents several moving vortex / bubble configurations. These all seem to be in a dipole configuration with windings (+1,-1) judging by the associated phase profiles. Is there something to be said concerning the individual (single) bubble / vortex solutions, i.e. density and phase plots corresponding to the data of Fig. 2? This might be connected to the numerical technique the authors have used “squared-operator iteration method” [page 5] (see further point in comment [vi] below) - a moving frame approach which produces excitations with oppositely signed pairs? Then, what are the physical properties of excitations with same-sign winding numbers (\pm1, \pm1) in this model, and what are the associated energies of the different configurations? This point should be clearly addressed.

(ii) The data presented in Fig. 4(d) is somewhat jagged compared to the other panels of this figure. Could the authors comment on why this is the case?

(iii) The authors state on page twelve “The individual states are taken as the obtained solutions with an additional 3% random perturbation” - as I understood the data presented in Fig. 6 onwards is in the presence of such noise - how do the presented results depend on how this quantity is chosen? What happens if no noise is present, do the bubbles still expand and do the authors know if this is linked to the metastability of the bubble states, as described in Fig. 1?

(iv) In the figures showing the dynamics of the bubbles, it would help clarify the results if the authors could compute the time-dependent root-mean-squared widths \sigma_{x,y} of the bubble’s density n = n_0 - |\psi|^2. The authors claim (page 13) “..in Fig. 8(a), where it is evident that the bubble maintains its structural integrity during the acceleration process.” I am not sure the presented data supports this conclusion (the bubble undergoes elliptical deformation) and a more quantitative approach would help clarify this. The norm of the bubble \int dr (n_0 - |\psi|^2) may also be an insightful measure in this regard.

(v) Coupling an impurity to the beyond-mean-field model is an interesting question to explore. It would add value to the manuscript if the authors could provide some additional data exploring the effect of varying the number of impurity atoms - how does this affect the bubbles density profile and dynamics? (the atom number of the impurity is not mentioned in the text)

(vi) Typos and corrections. Below various corrections and amendments are listed for the author’s convenience

  • the authors should clearly state the definition of, and difference between a bubble and quantum vortex; perhaps around Ref. [32] in the introduction.

  • on page three, the authors state the excitation spectrum [Eq. 4] of the model. Could the authors state the form of the trial wave function \Psi used to obtain this, mentioned in the sentence prior to this?

  • page four, line 95 “..denotes the additional force field due to rotation.” - this is perhaps better described as the centrifugal term I would think.

  • on page five, the authors state “..we restrict our attention to the fundamental vortex with topological charge S=1, as the results for higher charges are analogous to those of the fundamental case.” Is this statement justified? For example in the related cubic Schroedinger system S>1 charges are unstable, breaking apart into vortex anti-vortex pairs. Unless this situation is different for the beyond-mean-field case, perhaps the authors should remove this claim.

  • also on page five, the authors state that they use “..the squared-operator iteration method..” could the authors provide a few details as to what this entails, perhaps in a supporting appendix with appropriate reference(s)?

  • page six, line 160 “As previously introduced,..” > “As previously discussed,..”

  • page eight, line 190 “..it becomes clear that the..” > “.. there is a..”

  • In Sec. 3.3 the authors discuss “dispersion relations” of the bubbles. I think this would be more accurately described as the excitation’s energy and not its dispersion as claimed (see Eq. 8).

  • page thirteen, line 294 “..studies of interactions between bubbles, between bubbles and vortices,..” please clarify this sentence.

  • The authors may wish to add a reference to the recent experimental work of Cavicchioli et al., Phys. Rev. Lett. 134 093401 (2025) concerning quantum droplets in Bose mixtures. The authors state prior to Eq. (9) “Following the idea proposed in Ref. [40-42],..” the authors should also include the work of Clark which originally proposed modelling charged impurities in the Helium liquids using a model similar to that of Eq.(9); see R. Clark, Phys. Lett. 16 42 (1965).

Recommendation

Ask for major revision

---

## Round 1 · Referee Report · Anonymous (Referee 3) · 2025-12-15

Strengths

1-Well structure write-up
2-Timely topic of experimental relevance
3-Unaddressed states are identified and discussed in detail

Weaknesses

1-Missing physics explanations 2- Missing figure assignments in the relevant discussions making the text hard to follow

Report

In the present work the authors study the existence and dynamics of static and traveling quantum bubbles as well as vortices (focusing on singly quantized ones) by considering the 2D single component extended Gross-Pitaevskii (eGPE) model. By closely following the work of Ref. the work of Ref. Phys. Rev. A 110, 033317 (2024) they construct an effective potential model for the setup at hand that in contrast to earlier findings includes two additional force terms. Within this effective potential description they are able to identify a region where bubbles and vortices coexist.

Additionally, among others, the authors identify traveling bubbles and vortices by numerically solving the eGPE in a co-moving reference frame. They showcase that the exists a critical velocity where traveling bubbles develop in their phase a vortex-antivortex pair, namely they acquire a topological character. Finally, they consider the dynamical evolution, of the perturbed via random noise, waveforms showing, among others, that non-moving bubbles are unstable while they become more resilient for increasing velocities.

I find that the present work is well structured, timely and includes yet unaddressed states, their properties and. However, I find that further clarifications are needed especially around the role of the extra forces identified in the effective potential description. I also find that it would be beneficial for the work in terms of enhancing its readability if the authors could explicitly state the intervals of existence of all waveforms.

In view of the above, I cannot suggest for publication in SciPost Physics, this work in its present form, unless the authors are able to address and include in their work all points appended below.

Minor comments:

1) I would encourage the authors to check the entire manuscript for typos. For instance, there is a typo in line 71 midway, and another one appears e.g. in line 81.

Major comments:

1) I find the motivation provided currently in the introduction is not that strong. For instance, even though the states identified in the present work are new, why are they important? What are the potential applications of the setup at hand?

2) According to the effective potential analysis presented in Figure 1 of the manuscript, no coherent structures are present as dictated by U(\phi) (dashed black line), bubbles occur when two local minima are formed (red solid) but it is unclear how one can read off this effective potential the limiting case of an infinitely extended bubble. Can the authors clarify this further?

3) Along the aforementioned lines, the authors closely following the work of Ref. Phys. Rev. A 110, 033317 (2024) which they also cite indicate the absence of the radial kink. Can they provide further clarifications as to why the radial kink solution appears to be absent in their setup?

4) Since the background density, n_b, is directly connected to the chemical potential, and n_0 and n_s are constants (in the dimensionless units adopted by the authors), one can extract explicitly the regions of existence of the distinct solutions providing the actual numbers. I find that stating the regions of existence would not only help non expert readers but would also enhance the readability of the present work and would further provide connections/distinctions with earlier works. Therefore, I would strongly encourage the authors to include those bounds in the work.

5) Since the major deviation from earlier works in the field is directly related with the additional forces F_D and F_S, I find it essential for the authors to provide figure illustrating those forces or even better the deformation the cause in the effective potential.

6) Coexistence from fig.1 could be inferred for vortices and droplets not bubbles. So it is currently unclear how the deformation of the effective potential can alter the region of existence of vortices that much, so as to be able to coincide with bubbles. Can the authors comment on that and add a relevant discussion in the text?

7) In line with the above question, can the authors explicitly state how the obtain the critical density n*=0.95? Currently it is stated that one needs to take into account both F_D and F_S.

8) Can the authors explain how they obtain the traveling bubbles? Namely, what is the actual iteration scheme stated in the text that the authors used to numerically solve Eq. (7).

9) There is indeed a vortical phase structure shown e.g in Fig. 3(b_2 phase) but is there an imprint of it in the density of the bubble configuration? If yes this is hardly visible in the relevant figure, so please try to amend the figure so as to be visible and assign the relevant panels in the discussion to enhance the readability of the present work.

10) How do the authors extract the different velocities shown in Fig. 4 (c)? Please include a relevant discussion in the text.

11) Can the authors explicitly state how d is defined which is then presented as a function of velocity in Fig. 4(a)?

12) In Fig. 5 the authors present the excitation energy for traveling bubbles and vortices with the shading corresponding to stable (green) and unstable (pink) regions. Can the authors explain how do they obtain those and include a relevant discussion in the text? Additionally, assignment of figure panels around this discussion has to be included to make it more accessible.

13) In line with the above question, can the authors explain why the unstable region shown in Fig. 5(b) is no longer present for the higher background densities shown in Fig. 5(c)?

14) What is the difference in the ansatze used for the dynamical evolution of the vortices shown in Fig. 7?

Recommendation

Ask for major revision

---

## Editorial Decision

awaiting_resubmission